# Numerical and Experimental Research on Non-Reference Damage Localization Based on the Improved Two-Arrival-Time Difference Method

**DOI:** 10.3390/s22218432

**Published:** 2022-11-02

**Authors:** Yaogang Wu, Kangwei Liu, Dinghe Li, Xing Shen, Pengcheng Lu

**Affiliations:** 1College of Aeronautical Engineering, Civil Aviation University of China, Tianjin 300300, China; 2College of Aerospace Engineering, Nanjing University of Aeronautics and Astronautics, Nanjing 210016, China

**Keywords:** Lamb waves, piezoelectric ceramic lead zirconate titanate (PZT) transducers, non-reference damage localization (NRDL) method

## Abstract

The identification of damage based on Lamb waves can hardly avoid obtaining the reference signal under healthy conditions. A non-reference damage localization (NRDL) method is proposed in this paper. The NRDL method is established by the improved two-arrival-time difference method (2/ATDM) and BFGS method. The layout principles of the piezoelectric ceramic lead zirconate titanate (PZT) transducer array in the specimen are proposed. In contrast to existing methods, the damage outside the array in the specimen is identified based on the NRDL. The full-area damage location in the specimen is realized. Furthermore, the optimization of the excitation signal center frequency and transducer array layout is carried out. The damage location accuracy is greatly improved.

## 1. Introduction

With the rapid development of industry, large plate or plate-shaped metal engineering structures are widely used in the field of aviation, in components such as the fuselage and hatch. In the complex and changeable service environment and working conditions, these structures have sufficient safety and reliability. However, the accumulation of damage and the degradation of function will occur under the long-term influence of human factors and material aging. If the damage is not found in time, it will not only affect the normal use of the structure, but also lead to catastrophic accidents and heavy casualties in extreme cases. Damages cannot be directly identified by visual inspection due to the diversity of forms. With the development of non-destructive testing (NDT) technology (such as eddy current testing and ultrasonic testing), this problem has been alleviated to a certain extent. NDT technology is a form of passive monitoring. It can not immediately obtain the source and severity of damage. With the emergence of intelligent materials, structure health monitoring (SHM) technology has been rapidly developed. Active monitoring makes it possible to effectively estimate the remaining life of the structure. Among these approaches, an active detection technology consisting of Lamb waves based on Piezoelectric ceramic lead zirconate titanate (PZT) transducers is widely used by scholars because of its wide application range and high monitoring accuracy.

Two methods are usually used to analyze the arrival time of Lamb waves: the time domain method and time-frequency domain method. Kehlenbach et al. [1] proposed a method for damage identification based on an elliptic algorithm. It used the time of flight (ToF) to determine the damage localization in the structure. Baptista et al. [2] investigated the performance of time reversal for the localization of impact sources on fiber-reinforced plastic composite structures with embedded piezoelectric sensors. Furthermore, they then studied the feasibility of a piezoelectric diaphragm instead of piezoelectric ceramic and compared the effect of temperature variation on both sensors [3]. The variations in the amplitude, energy and cross-correlation of signals in health and damage states was researched by Burkov et al. [4]. Furthermore, the severity of the damage was obtained. Zhou et al. [5] used the rectangular actuator to excite axisymmetric and non-axisymmetric Lamb waves. They proposed an analytic expression to predict the amplitude of each circular ridge Lamb mode. The sensitivity is improved by selecting the appropriate rectangular actuator for damage detection. Perelli et al. [6] proposed a flat-plate acoustic emission localization strategy with dispersion and reverberation. The method extracted the differences in the distances traversed by stress-guided waves. The difficulties associated with the arrival time detection based on the classical thresholding procedure were overcome. Zhu et al. [7,8] evaluated fatigue damage by using the group velocities of non-linear Lamb waves. An effective mode was selected and a parameter was proposed to quantify the efficiency of the cumulative second harmonic generation (SHG) of Lamb waves. The results show that the S3-S6 modes were sensitive to the evolution of fatigue damage. The integrated amplitude of the second harmonic increased by nearly 300 percent with increasing fatigue period. The non-linear Lamb waves with group velocity mismatch were validated as candidates for effective fatigue damage assessment. Zhang et al. [9] proposed a method for detecting corrosion depth at the edges of holes by using the A0 mode of Lamb waves. Zhan et al. [10] focused on the non-linear response of two different frequencies of Lamb waves in a plate structure with micro-cracks. It was demonstrated that the mixed-frequency Lamb waves technique was sensitive to micro-cracks. The length of the cracks had a relationship with the harmonic amplitude. Rai and Mitra [11,12] proposed a deep learning structure based on a one-dimensional convolutional neural network, which could directly process the recorded original discrete time-domain signals of Lamb waves. The damage detection results were improved by a hybrid physics-assisted multi-layer feed-forward neural network model. Zonzini et al. [13] proposed a multi-layer convolutional neural network and a capsule structure with dedicated time retrieval logic to estimate the time of arrival of acoustic emission signals. It had the particular advantage of providing consistent results in the presence of significantly low signal-to-noise ratios. Ma and Yu [14] proposed quantifying the complex damage of plate structures by using spatiotemporal Lamb waves. The outline information of the damage was clearly obtained. Giurgiutiu et al. [15] applied guided Lamb waves to detect cracks in aluminum specimens. Three wave propagation methods were applied to detect three types of damage. Borate et al. [16] proposed a data-driven structural health monitoring method for the in situ assessment of the localization and extent of damage. Wang et al. [17] used an active sensing method based on Lamb waves and an improved sensor network to detect corrosion expansion at the hole edge. Some scholars have noticed the damage localization method without a baseline. Hameed et al. [18,19] proposed a damage estimation method based on the continuous wavelet transform of the normalized Lamb wave signal. The proposed method can estimate the localization and severity of multiple types of plate damage without baseline signals from undamaged plates. Zhou et al. [20] proposed a reference-free crack-detection method based on Lamb wave mode conversion. The Lamb wave mode conversion technique was adopted and the cracks could be quickly detected. Deng et al. [21] proposed a focused multiple signal classification (MUSIC) algorithm based on the virtual time reversal technique. The baseline free Lamb wave damage localization of isotropic materials was achieved. However, these methods cannot realize damage identification outside the transducer array. De Fenza et al. [22] proposed an astroid algorithm with the two-arrival-time difference method (2/ATDM). In our previous article, a damage shape recognition algorithm was proposed based on the 2/ATDM [23]. The layout and number of sensors have been studied by several scholars. Chen et al. [24] proposed an optimal load-dependent sensor arrangement scheme considering multi-source uncertainty. Furthermore, they [25] proposed an algorithm for structural health monitoring sensor layout based on a subclustering strategy. The algorithm is able to reduce redundancy and improve the performance of sensor configurations.

At present, damage identification based on Lamb waves mainly focuses on damage localization. There are also some studies focused on the imaging of different damage types or multiple damages. None of these efforts can avoid the existence of a problem. It is necessary to obtain the reference Lamb wave signal under healthy conditions. Healthy reference signals are not available in all monitoring processes. Moreover, the layouts of the PZT transducer array in the structure are usually required. Under the same monitoring requirements, the fewer transducers and more concise layout in the array can improve the ability to identify damage. Therefore, a non-reference damage localization (NRDL) method for aluminum alloy plate is presented in this paper. The actual aviation structure is complex, and not all sensors are suitable for deployment. If the monitoring range of the existing transducer array can be expanded, the application of SHM will be improved. Through the further study of the proposed NRDL method, the damage outside the array is also well-monitored in the specimen. The following parts of this article are organized as follows: In Section 2, the proposed NRDL method is presented with the detailed process of combining the improved 2/ATDM and BFGS methods. In Section 3, the basic theory of Lamb waves is introduced. The processes of the experiment are presented and the steps of the simulation are presented in as much detail as possible. The convergence of mesh size and the time increment step are considered. In Section 4, the analysis process of the NRDL method is proposed in detail. Based on NRDL, the numerical results are in good agreement with those of the experiment. Furthermore, the layout of the transducers and the center frequency of the excitation signal are optimized, and better damage localization results are obtained. Finally, some conclusions and limitations are drawn in Section 5.

## 2. Non-Reference Damage Localization Method

An NRDL method is proposed by improving the governing equation of 2/ATDM. The schematic diagram of the improved 2/ATDM is shown in Figure 1. PZT0 is used as the actuator. PZT1, PZT2 and PZT3 are taken as the sensors. When there is damage in the specimen, the signals received by PZT1, PZT2 and PZT3 contain the direct wave signal from the PZT0 and the reflected signal after the damage. When the damage is close to the boundary, the signal reflected by the boundary will also be received.

Let the time taken by the excitation signal to reach the sensor through damage be *t*. The distance of the arrived direct wave signal is *Z*. The distance of the damage reflection signal is *D*, and the distance of the possible boundary reflection signal is *R*; then, there is
(1)$$Z_{n}+D_{n}+R_{n}=c_{g}*t_{n}$$
where *n* is the sensor number n=1,2,3. $c_{g}$ is the group velocity of the Lamb waves. As shown in Figure 1, the distances from PZT0 to PZT1, PZT2 and PZT3 are the same, and $Z_{1}=Z_{2}=Z_{3}$. Depending on the layout of the actuator and sensor, the arrival time of the boundary reflection is consistent. Then, there exists $R_{1}=R_{2}=R_{3}$. Let the received signals of the three groups of sensors differ from each other. The governing equation of the improved 2/ATDM is given by
(2)$$\left\{\begin{array}{c} D_{1}-D_{2}=c_{g}*\left(t_{1}-t_{2}\right)\\ D_{2}-D_{3}=c_{g}*\left(t_{2}-t_{3}\right)\\ D_{3}-D_{1}=c_{g}*\left(t_{3}-t_{1}\right) \end{array}\right.$$
where $D_{1}=D\left(P_{0}d\right)+D\left(DP_{1}\right)$, $D_{2}=D\left(P_{0}d\right)+D\left(DP_{2}\right)$, $D_{3}=D\left(P_{0}d\right)+D\left(DP_{3}\right)$. $D(P_{0}d)$ denotes the distance from the actuator to the sensor. $D(DP_{n})$ is the distance from the damage to each sensor. Therefore, the above governing equation can be rewritten as
(3)$$\left\{\begin{array}{c} D\left(DP_{1}\right)-D\left(DP_{2}\right)=c_{g}*\left(t_{1}-t_{2}\right)\\ D\left(DP_{2}\right)-D\left(DP_{3}\right)=c_{g}*\left(t_{2}-t_{3}\right)\\ D\left(DP_{3}\right)-D\left(DP_{1}\right)=c_{g}*\left(t_{3}-t_{1}\right) \end{array}\right.$$

According to the Equation (Equation 3), any two sets of equations differ from each other. Three sensors can obtain three groups of hyperbolas. These hyperbolas focus on the position of any two sensors. Every two groups of hyperbolas will intersect at one point, and these intersections may be the localization where the damage occurs. Then, the accuracy of $t_{n}$ in the governing equation will seriously affect the final results of localization. The arrival time is easily affected by crosstalk and other factors. Let $t_{n}^{*}=t_{n}+t$, where *t* is the peak time of the excitation signal. Equation (Equation 3) is rewritten as
(4)$$\left\{\begin{array}{c} D\left(DP_{1}\right)-D\left(DP_{2}\right)=c_{g}*\left(t_{1}^{*}-t_{2}^{*}\right)\\ D\left(DP_{2}\right)-D\left(DP_{3}\right)=c_{g}*\left(t_{2}^{*}-t_{3}^{*}\right)\\ D\left(DP_{3}\right)-D\left(DP_{1}\right)=c_{g}*\left(t_{3}^{*}-t_{1}^{*}\right) \end{array}\right.$$

The continuous wavelet transform (CWT) is introduced to obtain the accurate $t_{n}^{*}$ [26,27]. The governing equation of CWT is given by
(5)$$W\left(u,s\right)=\frac{1}{\sqrt{s}}\int_{-\infty }^{\infty }f\left(t\right)\varphi^{*}\left(\frac{t-u}{s}\right)dt$$
where f(t) denotes the input signal. *s* denotes the number of decimal places. *u* is the position. W(u,s) is a soft function of wavelet coefficients f(t), and $\varphi^{*}\left(\frac{t-u}{s}\right)$ is a wavelet function in the complex domain.

Although multiple intersections are found, the localization of damage is still uncertain. A quasi-Newton optimization algorithm, the BFGS method, is introduced. The BFGS method was jointly studied by C.G. Broyden, R. Fletcher, D. Goldfarb, and D.F. Shanno in 1970. It is a method of symmetric positive definite iterative matrix [28]. The BFGS method has excellent numerical stability. It is necessary to introduce the quasi-Newton method before describing the BFGS method. For an objective function f(x) existing in the plane, make a vertical line through any point $x_{1}$ on the *x*-axis. The point $f(x_{1})$ is obtained, which is also a point on the objective function. Make an intersecting line to f(x) passing through $f(x_{1})$. The intersection point $x_{2}$ is obtained. The above steps are iterated repeatedly. The obtained $f(x_{n})$ becomes closer to the root of the objective function. However, the computing time also increases. It is necessary to determine a threshold such that when n=k, $x_{k+1}$ subtracted from $x_{k}$ is less than this threshold. $x_{k}$ is approximately considered the root of f(x). If f(x) is a differentiable function, then there is
(6)$$x_{k+1}=x_{k}-\frac{f(x_{k})}{f^{{}^{\prime }}\left(x_{k}\right)}$$

Based on the second-order Taylor expansion, the Equation (Equation 6) is rewritten as
(7)$$x_{k+1}=x_{k}-\frac{f^{{}^{\prime }}\left(x_{k}\right)}{f^{\prime \prime }\left(x_{k}\right)}$$

The optimized objective function f(x) is a multivariate function. The Hessian matrix is introduced and the equivalent matrix of Equation (Equation 7) is given by
(8)$$x_{k+1}=x_{k}-H_{k}^{-1}*g_{k},k=0,1\cdots$$
where $g_{k}$ is $f^{{}^{\prime }}\left(x\right)$ in Equation (Equation 7) and $H_{k}$ is the second derivative f(x). The BFGS method is an algorithm to find $H_{k}^{-1}$ by iteration. The governing equation of the BFGS method is given by
(9)$$D_{k+1}=\left(I-\frac{s_{k}y_{k}^{T}}{y_{K}^{T}s_{k}}\right)D_{k}\left(I-\frac{y_{k}s_{k}^{T}}{y_{k}^{T}s_{k}}\right)+\frac{s_{k}s_{k}^{T}}{y_{k}^{T}s_{k}}$$
where $D_{k}$ is $H_{k}^{-1}$, $s_{k}=x_{k+1}-x_{k}$, $y_{k}=g_{k+1}-g_{k}$, and $g_{k}$ is the derivative function of the original function. When k=1, $g_{k}$ is the unit matrix.

The objective function of the NRDL method is constructed. *j* intersections can be obtained through the improved 2/ATDM. The intersection is considered as valid in the specimen. The invalid intersection can be removed artificially. $\left(x_{i},y_{i}\right)\left(i=1,2,3\cdots j\right)$ is the *i*-th coordinate. $\left(x^{*},y^{*}\right)$ is the coordinate of the optimal point. The sum of the distance *P* from the optimal point to each intersection can be expressed as
(10)$$P=\sum_{i=1}^{j}\sqrt{{\left(x_{i}-x^{*}\right)}^{2}+{\left(y_{i}-y^{*}\right)}^{2}}$$

The objective function of the NRDL method can be defined as
(11)$$F\left(x,y\right)=min\left\{q\right\}=min\left\{\sum_{i=1}^{j}\sqrt{{\left(x_{i}-x^{*}\right)}^{2}+{\left(y_{i}-y^{*}\right)}^{2}}\right\}$$

## 3. Experiment and Simulation

### 3.1. Lamb Waves

In elastic solids, Lamb waves are excited in the form of elastic waves. The dispersion equation of Lamb waves is derived from elasticity [23].
(12)$$\left\{\begin{array}{c} \frac{tan(\beta d)}{tan(\alpha d)}=-\frac{4\varepsilon^{2}\alpha \beta }{{\left(\varepsilon^{2}-\beta^{2}\right)}^{2}}\\ \frac{tan(\beta d)}{tan(\alpha d)}=-\frac{{\left(\varepsilon^{2}-\beta^{2}\right)}^{2}}{4\varepsilon^{2}\alpha \beta } \end{array}\right.$$
where *d* denotes the thickness of the plate. In the variables $\alpha =\sqrt{\frac{\omega^{2}}{c_{l}^{2}}-k^{2}}$ and $\beta =\sqrt{\frac{\omega^{2}}{c_{t}^{2}}-k^{2}}$, ε denotes the wave number, and ω is the angular frequency of the resonant wave. *E*, υ, and ρ denote the Young’s modulus, Poisson’s ratio, and density of the plate, respectively. $c_{l}=\sqrt{\frac{E(1-\nu )}{\rho (1+\nu )(1-2\nu )}}$ and $c_{t}=\sqrt{\frac{E}{2\rho (1+\nu )}}$ are the propagation velocity of the plane stress longitudinal wave.

As can be seen from Equation (Equation 12), the Lamb wave has the characteristics of multiple modes and dispersion. When the thickness of the plate is 1 mm, the phase velocity dispersion curves of Lamb wave are shown in Figure 2a. The relationship between group velocity and phase velocity is given by
(13)$$c_{g}=c_{p}{\left[1-\frac{\omega }{c_{p}}\frac{\partial c_{p}}{\partial \omega }\right]}^{-1}$$

The group velocity dispersion curves are shown in Figure 2b. When the frequency is less than 100 kHz, only the S0 and A0 mode are generated. Although the wavelength of the A0 mode is larger than that of the S0 mode, the propagation speed of the S0 mode is greater, and can reduce the superposition of boundary reflection signals and damage signals. Therefore, the S0 mode is adopted.

### 3.2. Details of Experiment

To validate the NRDL method, an experiment with the S0 mode is conducted on a 600mm×600mm×1mm aluminum alloy plate. The material properties of the plate specimen are given in Table 1. Lamb waves are excited by a Keysight 33500b function generator. A Tektronix 3024 four-channel digital storage oscilloscope is used to receive the response signals. In experiment, the plate specimen is placed on a soft towel to simulate the free vibration. The experimental setup is shown in Figure 3.

The narrow-band excitation signal is a five-cycle sine burst centered at 350 kHz, as shown in Figure 4. The amplitude of the excitation signal is 10 V. Five PZT transducers (PZT0, PZT1, PZT2, PZT3 and PZT4) are arranged on the specimen according to the NRDL method.

Taking the lower left corner of the specimen as the coordinate origin, the coordinates of those transducers are (300,300), (200,200), (400,200), (400, 400) and (200, 400), respectively. These PZT transducers are selected by the D33 instrument and WK6500B low-frequency impedance analyzer, and their properties are almost the same. The diameter of these PZT transducers is 10 mm and the thickness is 0.5 mm. The properties are shown in Table 1. A 6 mm diameter hole damage is prefabricated. Generally, the PZT transducer array divides the specimen into three regions: A, B and C. The three regions are represented by green, yellow and blue, respectively. A schematic diagram is shown in Figure 5.

The prefabricated hole damage centers in the three regions are $D_{A}\left(220,360\right)$, $D_{B}\left(320,440\right)$ and $D_{C}\left(460,440\right)$, respectively. In order to distinguish the damages in the three regions, these damages are called Case I, Case II and Case III, as shown in Figure 6.

### 3.3. Numerical Simulation

The NRDL method is numerically studied in this paper. Only the finite element analysis model of Case I is given, as shown in Figure 7. The in-plane mesh size of the three-dimensional model is 1mm×1mm and that in the thickness direction is 0.5 mm. The hole damage and the surroundings of the piezoelectric sheet are encrypted. The SOLID5, SOLID185 and SOLID45 elements are employed to simulate the plate, PZT transducer and bonding material, respectively. The total numbers of nodes and elements of the proposed model are 715,488 and 1,082,965, respectively.

In our previous research, the requirements for the convergence of mesh size and time increment steps were given [23]. The mesh size is affected by the group velocity of the S0 mode. The time increment step is subject to the center frequency of the excitation signal. The convergence analysis results are shown in Figure 8.

Finally, as shown in Figure 8, the mesh size is 1 mm, and the time increment step is $1.5\times 10^{-7}$ s.

## 4. Results and Discussion

### 4.1. Validations

The responses of the Lamb waves in Case I, Case II and Case III are experimentally and numerically compared. The results are shown in Figure 9, Figure 10 and Figure 11, where (a–d) indicate the excitation signals received by PZT1, PZT2, PZT3 and PZT4 as sensors, respectively.

As shown in Figure 9, Figure 10 and Figure 11, the results of the simulation are in good agreement with those of the experiment in each case. Taking Case I as an example, the difference between the six groups of received signals is calculated. The six groups of difference signals are processed by CWT. The CWT coefficients of the numerical results are shown in Figure 12. In order to further effectively extract the information of the difference signal, three principles are followed,

(1) The same properties of the transducers make the direct arrival signals of S0 mode and A0 mode the same. The actuator and sensors are symmetrically laid in the specimen. The signals reflected from the boundary are almost the same. When damage occurs, the only difference between the signals received by each sensor is the reflected signal of the damage. In order to avoid the superimposed signal interfering with the extraction of the damage signal, the S0 mode with faster propagation speed is adopted as the signal for identifying the damage.

(2) Taking Case I (damage occurs in the PZT transducers array) as an example, there are two extremes in the localization of damage: (a) the damage completely overlaps with the position of the actuator PZT0; and (b) the damage overlaps with the position of any sensor (PZT1, PZT2, PZT3 and PZT4). For these two extremes, there are also two possibilities for the signal reflected by the damage: (a) when the damage completely overlaps with the position of the actuator PZT0, the signals received by all sensors are consistent. The arrival time of the excitation signal is $2.6\times 10^{-5}$ s. The $t_{i}^{*}$ in Equation (Equation 4) is $3.33\times 10^{-5}$ s; (b) when the damage overlaps with any sensor position, the longest arrival time of the excitation signal is $7.83\times 10^{-5}$ s. The $t_{i}^{*}$ in Equation (Equation 4) is $8.55\times 10^{-5}$ s. Therefore, the CWT coefficient of the difference signal is valid only when $3.33\times 10^{-5}\;s<t_{i}^{*}<8.55\times 10^{-5}$ s.

(3) The following conditions shall be met in order to prevent the differential signal from being interfered with:(14)$$\frac{D\left(DP_{i}\right)-D\left(DP_{j}\right)}{c_{g}}\geq T\left(i\neq j\right)$$
where the subscripts *i* and *j* denote the numbers of the two different piezoelectric sensors; $c_{g}$ denotes the group velocity of S0; and $D(DP_{i})$ and $D(DP_{j})$ are the distance between the damage and the different sensor, respectively. *T* denotes one period of the excitation signal. This can effectively ensure that the signals reflected by the damage are not superimposed. There is no interference between the acquired difference signals.

According to the principles of the proposed NRDL method, the times of the two different signal peaks reflected by the damage can be obtained from Figure 12a,c,e,f, respectively. Each set of time is brought into Equation (Equation 4). Multiple groups of hyperbolas are calculated and drawn. The number of sensors corresponding to the time of the damage signal can be clearly marked in Figure 12, because the hole damage is prefabricated. However, the number of sensors in actual monitoring cannot be determined. The proposed method requires all signals to be different from one another. The calculation result of Equation (Equation 4) is not affected. It is only necessary to calibrate the serial numbers of the sensor in a certain order. As shown in Figure 12b, two peaks are not found in the effective time range. This group of signals is discarded. As shown in Figure 12d, although the time of two peaks can be found within the set time range, the times are too close. The signal superposition may occur and the real time cannot be determined. Therefore, this group of signals is also discarded according to principle (3). Multiple groups of hyperbolas can be obtained on the basis of complying with the above principles, as shown in Figure 13.

As shown in Figure 13, seven intersections can be obtained. The coordinates of the intersections are shown in Table 2.

Similarly, the results of the experiment are also processed. The CWT coefficients are shown in Figure 14.

By extracting the time domain information of the effective CWT coefficient, the hyperbolas of the experimental results are obtained, as shown in Figure 15. Seven intersections are also obtained. The coordinates of the intersections are shown in Table 3.

The results in Table 2 and Table 3 are brought into the objective function F(x,y). The coordinates of the predicted damage localization center are shown in Table 4. The coordinates of the actual damage center are (220, 360).

As shown in Table 4, the localization of damage is accurately obtained in experiments and numerical simulations. This shows the effectiveness of the proposed method. However, there are still some errors in the experimental results from the perspective of absolute distance, for several reasons: (a) there may be some errors in the properties of PZT transducers in the experiment; (b) the existence of crosstalk makes some small disturbances unable to be eliminated by subtraction; and (c) the boundary of the prefabricated hole damage is not smooth. Usually, Lamb waves are reflected at the edge of the damage. A rough boundary will cause error, but this error may be positive. According to the above process, the damages in Case II and Case III are predicted. The coordinates of the damage center in Case II and Case III are (320, 440) and (460, 440), respectively. In these two cases, the predictions of damage localization are shown in Table 5.

As shown in Table 4 and Table 5, there is a large error in damage localization in Case III. The excitation signal takes a long time to reach the sensors, and is easily superimposed with the reflected signal of the boundary. The proposed method needs to be optimized. The intersections of curves determined by the NRDL method are the possible locations of the damage. If more curves can be obtained, more possible positions of the damage can be obtained. More accurate damage prediction can be obtained by introducing the BFGS method. According to the principle of the proposed method, the relationship between the difference signal and the period of the excitation signal needs to be considered. Two optimization methods are proposed: (a) frequency optimization, that is, the reason for choosing 350 kHz as the center frequency; and (b) the layout of eight PZT transducers is adopted.

### 4.2. Frequency Optimization

Taking Case I as an example, four groups of sinusoidal modulated signals with center frequencies of 200 kHz, 250 kHz, 300 kHz and 350 kHz are selected as excitation signals, respectively. According to the four center frequencies, the corresponding *T* in Equation (Equation 14) is calculated as $2.5\times 10^{-5}$ s, $2\times 10^{-5}$ s, $1.67\times 10^{-5}$ s and $1.428\times 10^{-5}$ s respectively. The numerical results of received signals with different center frequencies (200 kHz, 250 kHz and 300 kHz) are compared with the experimental results, as shown in Figure 16, Figure 17 and Figure 18.

As shown in Figure 16, Figure 17 and Figure 18, the results of the simulations are in good agreement with those of the experiment. The signal processed by CWT is compared with the actual damage localization. When the center frequency is 200 kHz, only two groups of difference signals (PZT2-PZT4 and PZT3-PZT4) can be obtained. When the center frequency is 250 kHz and 300 kHz, three groups of difference signals (PZT1-PZT4, PZT2-PZT4 and PZT3-PZT4) can be obtained. Based on the above time-domain information, the predictions of damage center coordinates are obtained, as shown in Table 6. The absolute distance between the predicted damage center and the actual damage center is also given, as shown in Table 7.

According to Table 6 and Table 7, when the center frequency of the excitation signal is 200 kHz, only two curves can be obtained. Only one intersection cannot be optimized by the BFGS algorithm. The intersection of these two curves will be considered as the predicted damage center. This obviously shows that the error is large in comparison with the results under other central frequencies. In particular, the damage location of the experiment is far beyond the actual damage area. The absolute distance is 17.5 times that of the hole radius. When the center frequency of the excitation signal is 250 kHz and 300 kHz, three hyperbolas can be obtained. Three hyperbolas intersect at one point in pairs. The prediction of damage localization can be obtained by the BFGS method. Since the number of intersections is the same, the two groups of results have the same number of differential signals. The prediction error is basically the same. The absolute distances in the two groups of experiments are about 4.5 times that of the hole radius. Therefore, the excitation signal with a central frequency of 350 kHz is finally adopted. Four groups of differential signals can be obtained. The prediction of the damage localization is more accurate. A large center frequency is not necessarily better. When the center frequency increases, the sampling points used for the excitation function need to be increased significantly. The received signal is prone to distortion. A valid ToF may not be obtained, which puts forward high requirements for the sampling frequency of the experiment. The amount of calculation increases exponentially for the simulation. Therefore, the most reasonable center frequency shall be determined.

### 4.3. Layout Optimization

According to the governing equation of the proposed NRDL method, the subtraction between the received signals can filter out the direct waves of the excitation signal. PZT transducers with the same distance from the actuator in the specimen are arranged. More response results for differential signals can be obtained. Therefore, based on the previous layout, a circle is drawn with the original actuator as the center and the distance between the actuator and the sensor as the radius. New PZT transducers are added in the vertical and horizontal directions on the circle, respectively. Taking Case I as an example, the new layout of the sensor is shown in Figure 19a.

The transmission of Lamb waves is affected by the form of meshes in the numerical model. The mesh of the central radiation is adopted to ensure that it is the same from the actuator to each sensor, as shown in Figure 19b. The numbers of elements and nodes of the new model are 630,650 and 958,955, respectively. The calculation scale is reduced compared with the previous model. The number of sensors is increased to eight. The original red PZT transducers (PZT1, PZT3, PZT5 and PZT7) are taken as one group, and the new blue PZT transducers (PZT2, PZT4, PZT6 and PZT8) are taken as another group. A new set of hyperbolas can be drawn by another four sensors. The hyperbolas of the experiment and simulation are shown in Figure 20.

As shown in Figure 20, a total of 32 intersections are obtained from the experimental hyperbola and 34 intersections are obtained from the numerical hyperbola. The coordinate values of the corresponding intersections are shown in Table 8 and Table 9, respectively. These intersections can be used to obtain a new prediction of damage localization, as shown in Table 10. After the optimization of the layout, the absolute distance between the center of the predicted damage localization and the actual center is significantly reduced.

The number of sensors was increased to eight. More hyperbolas can be obtained by using the received signals of all sensors. The prediction of damage localization through the intersection obtained by eight sensors is shown in Table 11.

The prediction of a group of eight sensors is more accurate than that of a group of four sensors. However, it is no better than the prediction of two groups of four sensors. The global error increases due to some intersections with large errors. The amount of calculation using the BFGS method also increases. Hence, the layout optimization of two groups of four sensors is used to predict the damage in Case II and Case III. The optimized prediction is shown in Table 12.

Compared with Table 4 and Table 5, the identification of two groups of four sensors has been greatly improved. In Case I, it is obvious that the absolute distances are greatly reduced in the experiment and simulation, and are far less than the radius of hole damage. In particular, the experimental results show that more intersections are obtained near the actual center of the damage. The local error caused by individual points is reduced. This also occurs in the numerical results of Case III. Similarly, the experimental results of Case III are optimized to some extent. However, the influence of the experimental environment and the insufficient number of intersections lead to the limitation of optimization, which also puts forward new challenges for further optimization.

## 5. Conclusions

A numerical simulation model of an aluminum alloy plate specimen with hole damage is proposed based on the Lamb waves technique, which can be used to determine the localization of the damage without healthy reference signals. The NRDL method is proposed based on the improved 2/ATDM and BFGS methods. The numerical results of the NRDL method are verified by those of the experiment. Two optimization models based on frequency improvement and layout improvement are established. From the conducted numerical and experimental investigations, the following main conclusions emerge:The damage location without a reference is realized based on the proposed method. The error of the damage location is effectively reduced by the optimization of the center frequency and array layout. This is of great significance for the application of practical engineering.The full-area damage localization is realized in the specimen. The damage identification range of the PZT transducer array is improved. The accuracy of damage identification is higher in Region B outside the array than that in Region C. Furthermore, damage shape identification should use more points in the array and Region B.The principles for extracting good time-frequency domain information are established. These principles can be used to effectively obtain the characteristics of the response signal. The influence of the signal superposition is reduced.

Some limitations of the proposed method are as follows:Small differences in the properties of the PZT transducers will lead to errors in the results during the experiment, which puts forward higher requirements for the manufacture and selection of PZT transducers.The proposed method requires that the distance between each sensor and the actuator is consistent. The accuracy of mesh discretization is high in numerical simulation.If the prefabricated damage is not smooth during the experiment, the damage reflection signal and A0 signal may be superimposed, so the damage signal cannot be effectively separated.

## Figures and Tables

**Figure 1 sensors-22-08432-f001:**
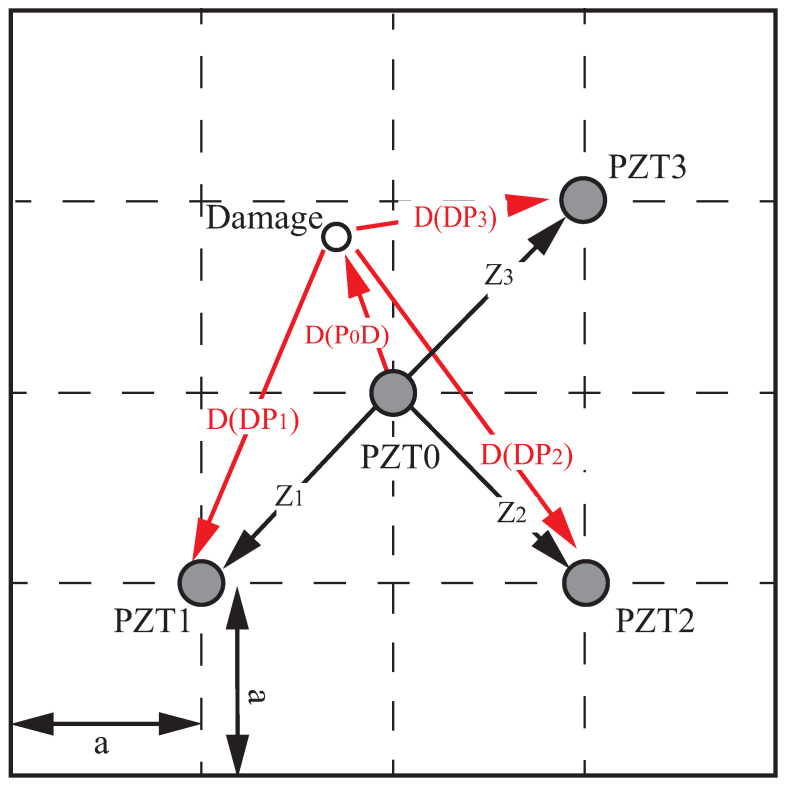
The schematic of the improved 2/ATDM.

**Figure 2 sensors-22-08432-f002:**
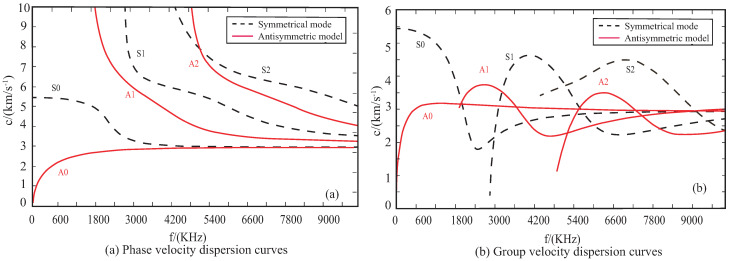
Dispersion curve of aluminum alloy plate.

**Figure 3 sensors-22-08432-f003:**
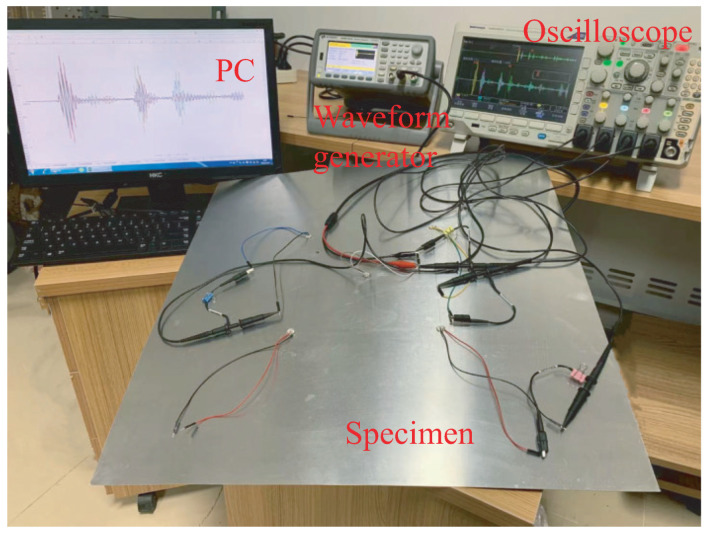
Experimental setup.

**Figure 4 sensors-22-08432-f004:**
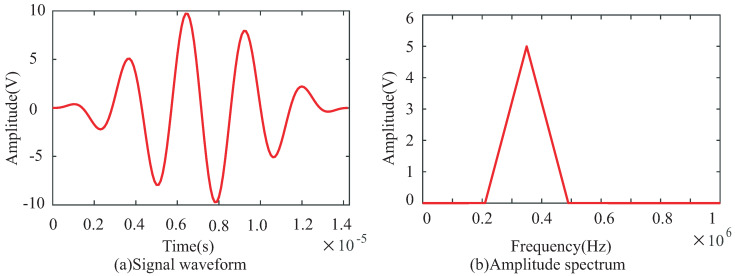
Narrow-band excitation signal with a central frequency of 350 kHz.

**Figure 5 sensors-22-08432-f005:**
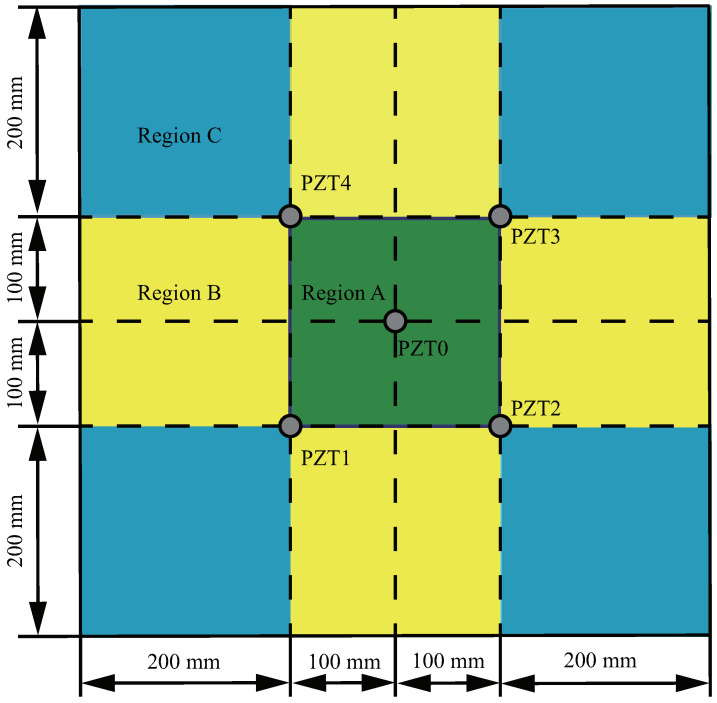
The schematic of specimen area division.

**Figure 6 sensors-22-08432-f006:**
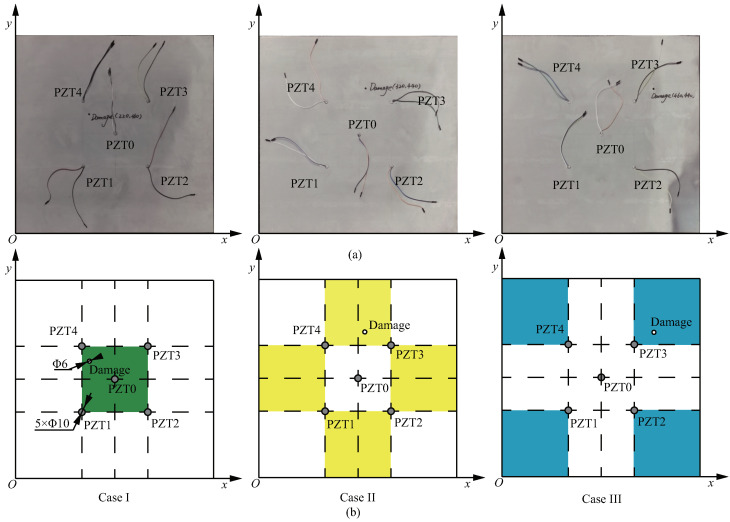
(**a**) Details of specimen, and (**b**) schematic.

**Figure 7 sensors-22-08432-f007:**
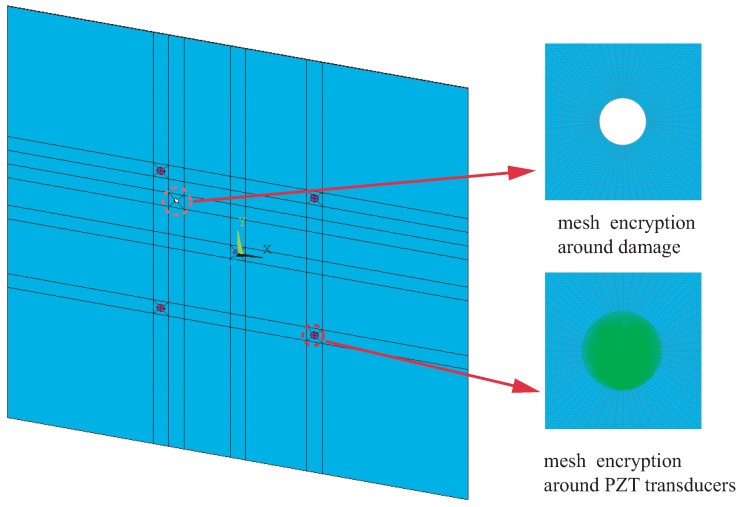
Details of numerical model.

**Figure 8 sensors-22-08432-f008:**
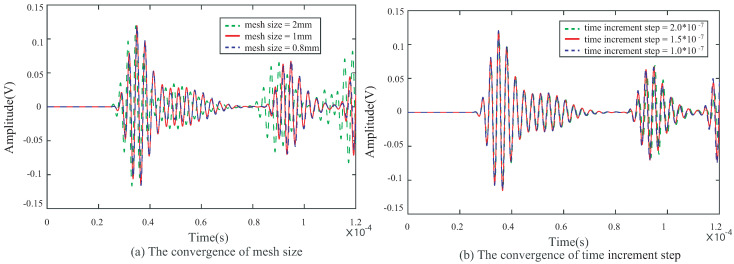
Results of convergence analysis: (**a**) mesh size and (**b**) time increment step.

**Figure 9 sensors-22-08432-f009:**
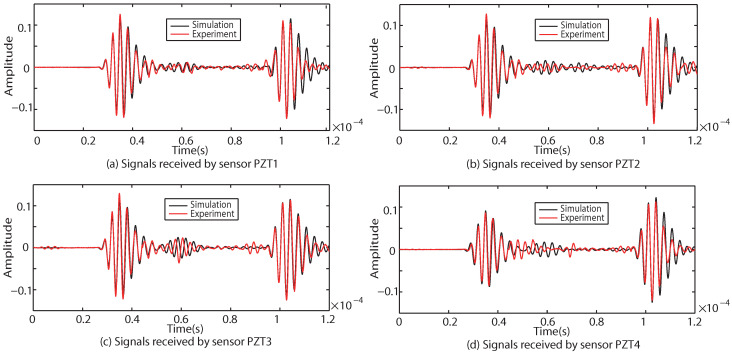
Comparison of experimental and simulation results under Case I.

**Figure 10 sensors-22-08432-f010:**
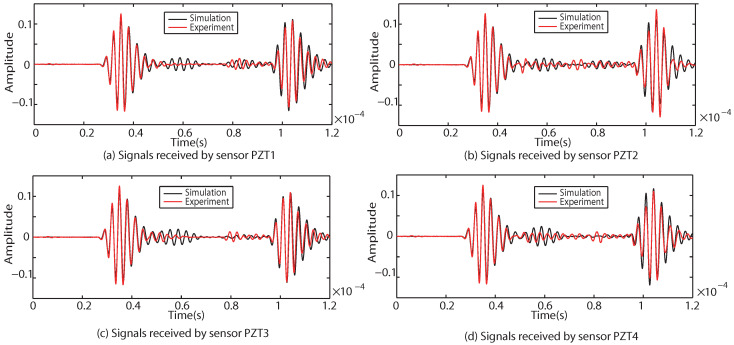
Comparison of experimental and simulation results under Case II.

**Figure 11 sensors-22-08432-f011:**
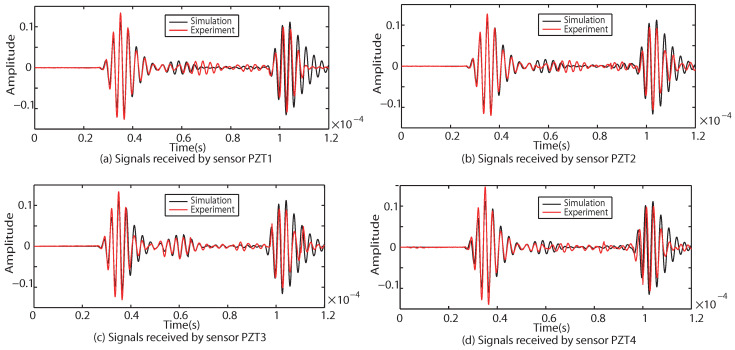
Comparison of experimental and simulation results under Case III.

**Figure 12 sensors-22-08432-f012:**
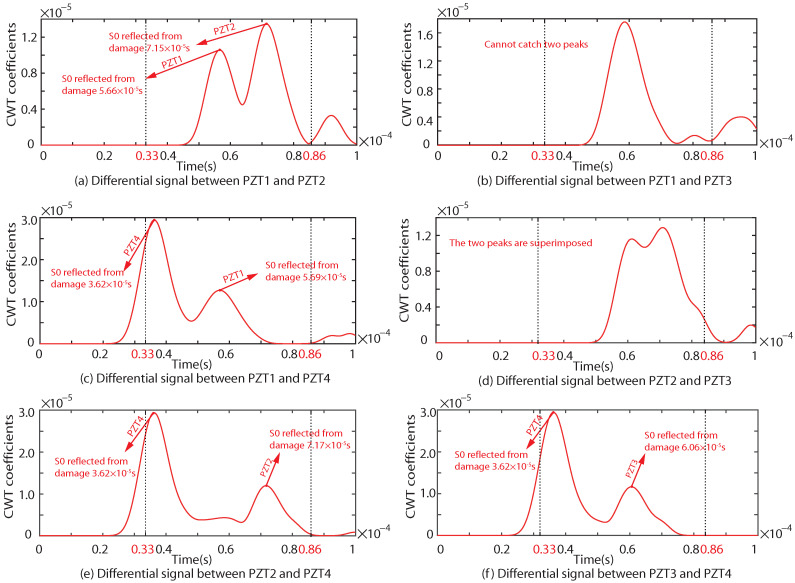
CWT coefficient of the difference signal of any two sensors under the simulation of Case I.

**Figure 13 sensors-22-08432-f013:**
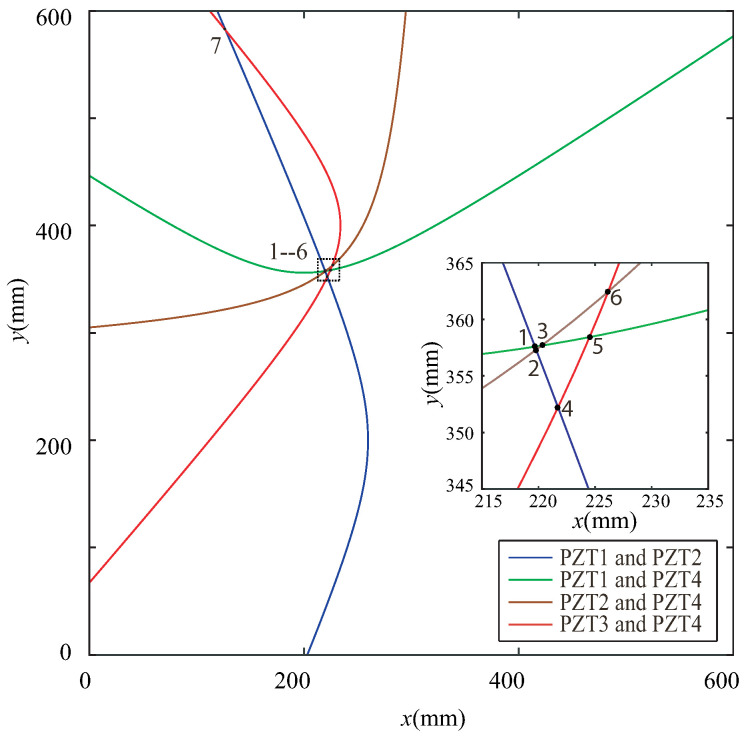
Intersection of simulations under Case I.

**Figure 14 sensors-22-08432-f014:**
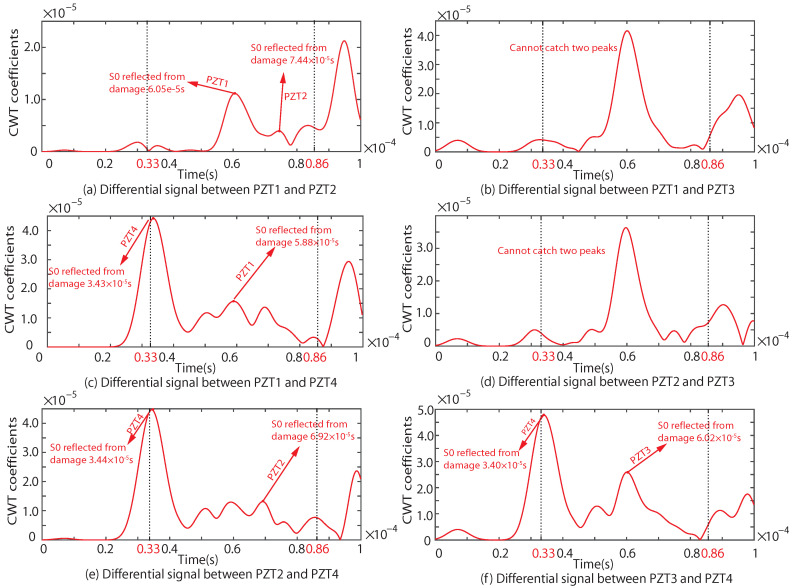
CWT coefficient of the difference signal of any two sensors in the experiment under Case I.

**Figure 15 sensors-22-08432-f015:**
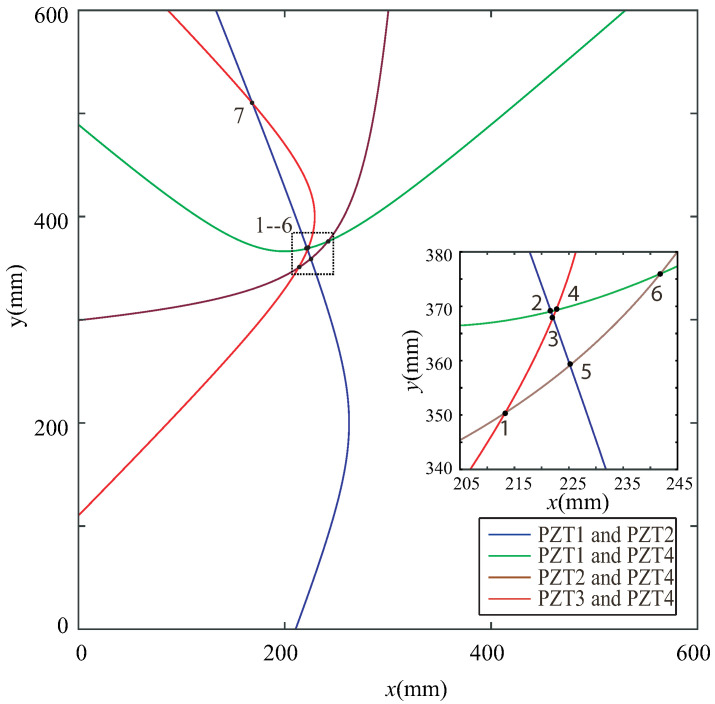
Intersection of experiment for Case I.

**Figure 16 sensors-22-08432-f016:**
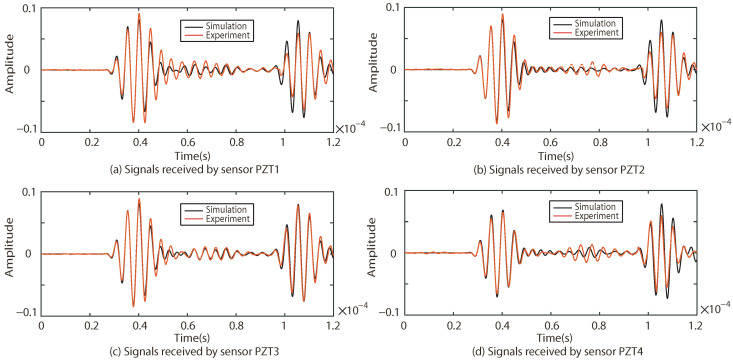
Comparison of experimental and simulation results under the center frequency of 200 kHz.

**Figure 17 sensors-22-08432-f017:**
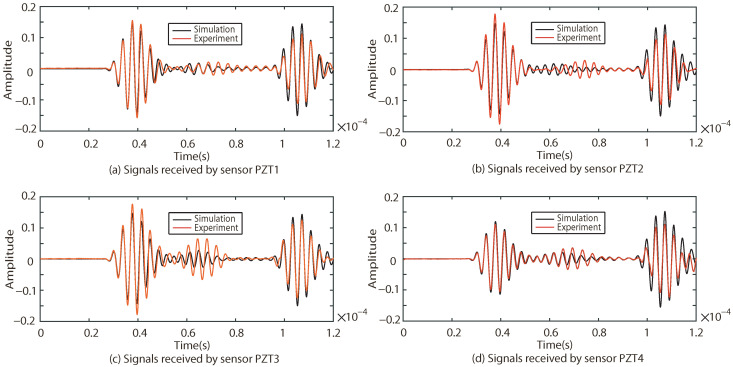
Comparison of experimental and simulation results under the center frequency of 250 kHz.

**Figure 18 sensors-22-08432-f018:**
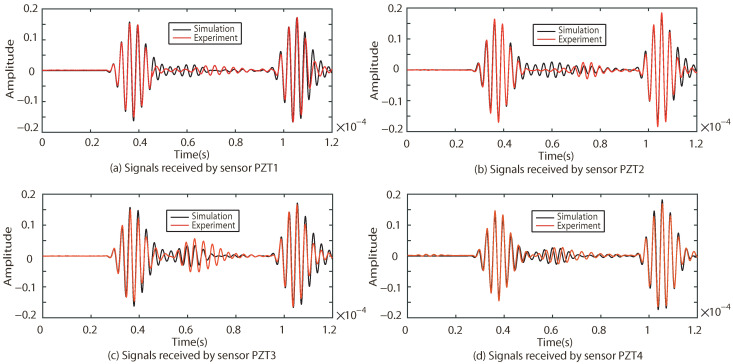
Comparison of experimental and simulation results under the center frequency of 300 kHz.

**Figure 19 sensors-22-08432-f019:**
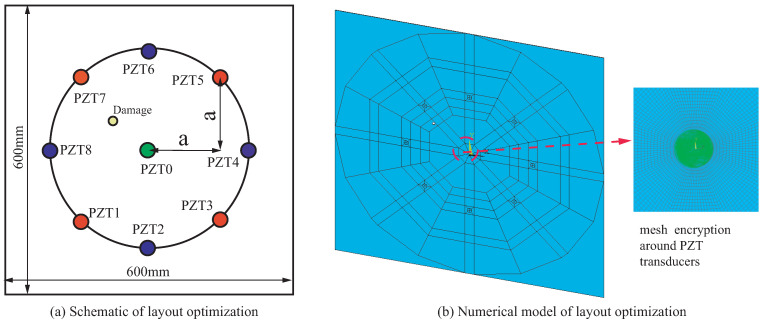
(**a**) The schematic of layout optimization, and (**b**) the numerical model.

**Figure 20 sensors-22-08432-f020:**
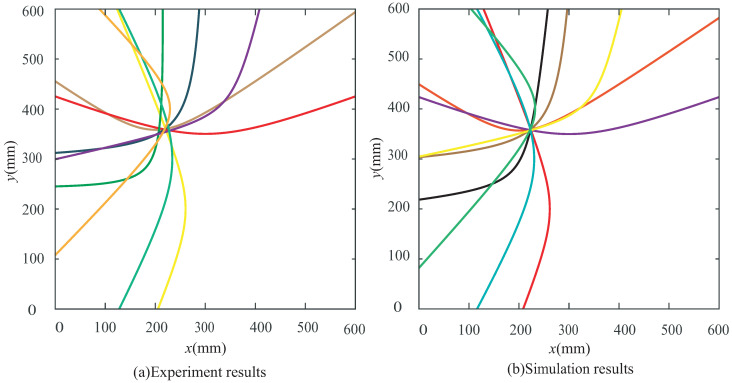
Hyperbolas of two groups of 4 sensors under Case I: (**a**) experiment and (**b**) simulation.

**Table 1 sensors-22-08432-t001:** Material properties of plate specimen and PZT transducers.

Properties of the Plate Specimen	
Young’s modulus (GPa)	67
Poisson’s ratio	0.34
Mass density $\left(\mathrm{kg}/m^{3}\right)$	2730
Properties of the PZT transducers	
Young’s modulus (GPa)	12.8
Piezoelectric constant $\left(c/m^{2}\right)$	−5.3
Relative permittivity $\left(\varepsilon_{33}\right)$	1000
Mass density $\left(\mathrm{kg}/m^{3}\right)$	7500

**Table 2 sensors-22-08432-t002:** Coordinates of the intersections under simulation.

	Point 1	Point 2	Point 3	Point 4	Point 5	Point 6	Point 7
x	219.6936	219.7961	221.7283	220.275	224.5164	226.1663	126.9352
y	357.5862	357.3172	352.2234	357.6784	358.427	362.453	581.902

**Table 3 sensors-22-08432-t003:** Coordinates of the intersections under experiment.

	Point 1	Point 2	Point 3	Point 4	Point 5	Point 6	Point 7
x	213.2933	221.7281	222.1135	222.6515	225.2257	241.8125	168.6522
y	350.3837	369.0463	367.9631	369.2788	359.1532	375.9530	509.9387

**Table 4 sensors-22-08432-t004:** Comparison of experiment and simulation results under Case I.

	Experimental Results				Numerical Results			
	**Coordinate**		**Error**	**Absolute Distance**	**Coordinate**		**Error**	**Absolute Distance**
Case I	x	222.2426	1.02%	8.8625	x	220.2751	0.13%	2.3377
	y	368.5741	2.38%		y	357.6785	0.64%	

**Table 5 sensors-22-08432-t005:** Comparison of experiment and simulation results under Case II and Case III.

	Experimental Results				Numerical Results			
	**Coordinate**		**Error**	**Absolute Distance**	**Coordinate**		**Error**	**Absolute Distance**
Case II	x	318.9059	0.256%	2.2472	x	319.3925	0.189%	3.5082
	y	438.0371	0.117%		y	436.5447	0.785%	
Case III	x	449.1647	2.35%	10.8504	x	447.4572	2.73%	12.8374
	y	439.4273	0.130%		y	437.2656	0.621%	

**Table 6 sensors-22-08432-t006:** Comparison of experimental and simulation results under different center frequencies.

Center Frequency	Experimental Results			Numerical Results		
	**Coordinate**		**Error**	**Coordinate**		**Error**
200 kHz	x	159.7300	27.40%	x	223.3914	1.54%
	y	323.4016	10.17%	y	361.6055	0.45%
250 kHz	x	235.6826	7.13%	x	222.3510	1.07%
	y	371.2715	3.13%	y	359.0424	0.27%
300 kHz	x	218.8626	0.52%	x	222.2883	1.04%
	y	341.8812	5.03%	y	358.6738	0.37%

**Table 7 sensors-22-08432-t007:** Absolute distance of experiment and simulation under different center frequencies.

	200 kHz	250 kHz	300 kHz	350 kHz
Numerical results	3.7522	2.5385	2.6448	2.3377
Experimental results	70.5118	19.3130	18.1545	8.8625

**Table 8 sensors-22-08432-t008:** Coordinates of the experimental intersections of two groups of 4 sensors under Case I.

Point	Coordinates		Point	Coordinates	
	* **x** * **(mm)**	* **y** * **(mm)**		* **x** * **(mm)**	* **y** * **(mm)**
1	221.4553	364.5463	17	223.3166	364.9534
2	224.0102	357.4715	18	222.7508	357.7103
3	223.2137	359.6855	19	222.6048	357.7382
4	223.6227	358.5495	20	224.0965	357.4553
5	224.0238	357.4338	21	223.1458	357.6351
6	221.2460	365.1226	22	220.2911	358.1851
7	181.7809	468.5540	23	222.9364	358.4981
8	222.2292	362.4113	24	221.1110	356.5792
9	156.6495	531.7421	25	223.0379	358.0812
10	201.0410	362.2550	26	218.5730	354.6916
11	140.8030	378.0512	27	223.2491	357.2055
12	224.4695	365.2212	28	219.2470	356.0378
13	233.8489	367.8223	29	222.0560	362.0188
14	350.6063	434.9999	30	143.4028	547.3921
15	221.4017	364.5351	31	297.7728	294.1370
16	230.9606	398.1050	32	297.6905	294.3483

**Table 9 sensors-22-08432-t009:** Coordinates of the simulation intersections of two groups of 4 sensors under Case I.

Point	Coordinates		Point	Coordinates	
	* **x** * **(mm)**	* **y** * **(mm)**		* **x** * **(mm)**	* **y** * **(mm)**
1	220.5253	360.3022	18	229.7778	357.0491
2	221.1313	358.6765	19	227.2649	357.5072
3	210.3383	387.1058	20	218.4681	359.2059
4	220.3237	360.8423	21	206.4816	351.6225
5	222.5655	354.8121	22	206.2354	349.9729
6	232.3564	327.6327	23	211.8637	410.1102
7	219.8613	362.0789	24	200.2313	348.2598
8	152.9875	529.1883	25	93.8966	321.1719
9	216.3509	359.6358	26	225.4863	365.1087
10	105.3407	389.8327	27	218.6479	359.5679
11	207.4560	358.7067	28	227.5217	356.3419
12	219.3574	360.1013	29	214.9374	352.5160
13	337.1930	414.8481	30	223.8542	371.5136
14	226.3602	361.4686	31	171.8906	507.0888
15	218.8712	360.0210	32	213.5070	450.0110
16	207.8005	361.4499	33	203.3735	333.9378
17	218.2258	359.2547	34	143.9420	260.5652

**Table 10 sensors-22-08432-t010:** Experimental and simulation results of two groups of 4 sensors under Case I.

	Experimental Results				Numerical Results			
	**Coordinate**		**Error**	**Absolute Distance**	**Coordinate**		**Error**	**Absolute Distance**
Case I	x	218.8806	0.51%	1.1768	x	222.3815	1.080%	2.7567
	y	360.3631	0.10%		y	358.6115	0.385%	

**Table 11 sensors-22-08432-t011:** Experimental and simulation results of 8 sensors under Case I.

	Experimental Results				Numerical Results			
	**Coordinate**		**Error**	**Absolute Distance**	**Coordinate**		**Error**	**Absolute Distance**
Case I	x	218.9052	0.50%	1.1648	x	222.8893	1.31%	2.5595
	y	359.6022	0.111%		y	358.4227	0.438%	

**Table 12 sensors-22-08432-t012:** Experimental and simulation results of two groups of 4 sensors under Case II and Case III.

	Experiment Results				Numerical Results			
	**Coordinate**		**Error**	**Absolute Distance**	**Coordinate**		**Error**	**Absolute Distance**
Case II	x	320.3661	0.475%	2.1066	x	319.0008	0.3123%	2.9088
	y	437.9254	0.080%		y	437.2682	0.621%	
Case III	x	467.0655	1.53%	7.7949	x	460.4487	0.097%	2.9586
	y	443.2924	0.748%		y	437.0756	0.664%	

## Data Availability

The data presented in this study is available on request from the corresponding author.

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
