# Peer review of "Numerical and Experimental Research on Non-Reference Damage Localization Based on the Improved Two-Arrival-Time Difference Method"

_sensors, 2022, doi:10.3390/s22218432_

Round 1
Reviewer 1 Report
This paper proposed a non-reference damage localisation (NRDL) method, which is established by the improved two arrival time difference method (2/ATDM) and BFGS method. The study is interesting with good organization. Some points should be revised before acceptance:
1. How does this method detect damage if the damage location is outside the sensor array.
2. The types of damage to which this method is applicable need to be introduced
3. How to determine the optimal number and location of sensors? Some references can be give some suggestions: “A novel load-dependent sensor placement method ……”, and “Sensor placement algorithm …… sub-clustering strategy”.
4. The impact of damage degree and noise on the accuracy of this method should be revealed.
5. The boundary conditions of Fig. 3 should be given.
Reviewer 2 Report
The reference for this article is low. There needs to be a lot more reference to competing numerical methods and how their method is superior. Also there has been no reference to other NRDL methods and investigating other research and how their work is superior to others is needed.
Reviewer 3 Report
This paper presents a method based on optimization based on non-reference damage
localization. The paper is well-written and easy to follow. Here are some points that need to improve.
1. Abstract should be compress give the major contribution
2. Section 4.3 should be discussed more
3. The discussion and analysis of Table 7 are not clear,
4. The Conclusion is short.
